# Life-course blood pressure trajectories and cardiovascular diseases: A population-based cohort study in China

Yongshi Xu[1], Jette Möller[1], Rui Wang[2,3,4], Yajun Liang[1]*

1 Department of Global Public Health, Karolinska Institutet, Stockholm, Sweden, 2 The Swedish School of Sport and Health Sciences, GIH, Stockholm, Sweden, 3 Division of Clinical Geriatrics, Department of Neurobiology, Care Sciences and Society, Karolinska Institutet and Stockholm University, Stockholm, Sweden, 4 Wisconsin Alzheimer's Disease Research Center, University of Wisconsin School of Medicine and Public Health, Madison, Wisconsin, United States of America

* yajun.liang@ki.se

## Abstract

### Background

The patterns of blood pressure trajectory (i.e., change over time) over life-course remain to be explored. In this study, we aim to determine the trajectories of systolic blood pressure (SBP) from adulthood to late life and to assess its impact on the risk of cardiovascular diseases (CVDs).

### Methods

Based on the China Health and Nutrition Survey, a total of 3566 participants aged 20–50 years at baseline (1989) with at least three SBP measurements during 1989–2011 were included. SBP was measured through physical examination, and socio-demographic factors, lifestyles, medications, and CVDs were based on self-reported questionnaire. Latent class growth modeling was performed to examine SBP trajectory. Odds ratio (OR) and 95% confidence interval (CI) from logistic regression was used to determine the association between SBP trajectory and CVDs.

### Results

Five trajectory groups of SBP were identified: Class 1: rapid increase (n = 113, 3.2%); Class 2: slight increase (n = 1958, 54.9%); Class 3: stable (n = 614, 17.2%); Class 4: increase (n = 800, 22.4%); Class 5: fluctuant (n = 81, 2.3%). After adjustment of demographic factors, baseline SBP, and lifestyles, compared with the "slight increase" group, the OR (95% CI) of CVDs was 0.65 (0.32, 1.28) for "stable" group, 2.24 (1.40, 3.58) for "increase" group, 3.95 (1.81, 8.62) for "rapid increase" group, and 4.32 (1.76, 10.57) for "fluctuant" group. After stratified by use of antihypertensive drugs, the association was only significant for "rapid increase" group among those using antihypertensive drugs with OR (95% CI) of 2.81 (1.01, 7.77).

**Data Availability Statement:** Data are from the CHNS project, an ongoing population-based longitudinal study (https://www.cpc.unc.edu/projects/china). Access to these original data is

available to the research community upon approval by the CHNS data management and maintenance committee. Applications for accessing these data can be submitted to the committee through filling in an online registration form (https://www.cpc.unc.edu/projects/china/data/datasets/data-downloads-registration). The authors had no special access privileges to the data that others would not have.

**Funding:** The China Health and Nutrition Survey (CHNS) was funded by a number of organizations. Main funding for the survey and data dissemination from 1991 to 2004 came from the National Institutes of Health (NIH) (P01-HD28076 and HD30880). Additional funding has come from NIH (HD39183), the Carolina Population Center (CPC) (in particular, CPC funded CHNS 1989), the Ford Foundation, the National Science Foundation (INT-9215399), the National Institute of Nutrition and Food Safety (formerly named Institute of Nutrition and Food Hygiene), and the Chinese Centers for Disease Control and Prevention (formerly named Chinese Academy of Preventive Medicine). This work was supported in part by the Karolinska Institutet, Sweden (2018-01590). The funders had no role in study design, data collection and analysis, decision to publish, or preparation of the manuscript.

**Competing interests:** The authors have declared that no competing interests exist.

## Conclusions

Having a rapidly increasing SBP over life-course is associated with a higher risk of CVDs. This implies the importance of monitoring lifetime change of blood pressure for the prevention of CVDs.

## Introduction

Previous studies on the association between high blood pressure and cardiovascular diseases (CVDs) have normally measured blood pressure at one occasion [1–3] or several occasions in a specific life period [4, 5]. However, the blood pressure of each individual varies over different life periods, and one-time value of blood pressure without considering the change of blood pressure levels is less reliable [6]. In addition to actual blood pressure level, the over-time changes of blood pressure (i.e., blood pressure trajectory) should be used to assess the risk of CVDs. Therefore, monitoring the level and trajectory of blood pressure is an essential part of the guidelines for CVD prevention [7, 8].

So far, several studies have assessed the trajectories of blood pressure and examined the association between blood pressure trajectory pattern and the risk of CVDs [6, 9–14]. However, the previous studies on blood pressure trajectories were confined to the change of blood pressure in a certain short period of life, e.g., only in childhood [9], from childhood to adulthood [6, 10], only in adulthood and midlife [11, 12], from midlife to late life [13], or only in late life [14]. There is little evidence about the patterns of blood pressure trajectory over life-course and their impacts on CVDs.

In addition, the previous studies on blood pressure trajectory focused on only systolic blood pressure (SBP) [6, 13], SBP and diastolic blood pressure (DBP) separately [9–12, 14] or in combinations [7, 11, 12, 14]. Some found that SBP trajectory was associated with atherosclerotic biomarkers (e.g., carotid intima–media thickness and left ventricular mass index) where DBP was not [10]. Others found that DBP trajectory groups were less strongly associated with coronary artery atherosclerosis than SBP trajectories [11]. In addition, SBP has been shown to be a better and stronger predictor than other blood pressure components for the risk of cardiovascular events [15, 16]. In regard to the above evidence, we focused only on SBP in this study.

Based on the longitudinal cohort from the China Health and Nutrition Survey (CHNS), this study aimed to examine the trajectory of SBP from young adulthood to old age as well as to determine the association between distinct patterns and CVDs.

## Methods

### Study design and population

Data from the CHNS was retrieved for this study. The details of CHNS have been described elsewhere [17]. In short, the CHNS is an ongoing cohort study, which was initiated in 1989 with follow-ups in every 2 to 4 years, aiming to assess the health status and risk factors of the Chinese population. Participants were recruited from fifteen provinces and municipal cities in China through a stratified cluster randomization procedure. In CHNS, data were collected by the well-trained examiners through questionnaire investigation, physical examination, and laboratory tests according to a consistent protocol in each wave [17].

For the purpose of this study, 3566 participants (female, 52.0%) aged from 20 to 50 years at baseline (mean age = 33.3 years, standard deviation = 6.8 years) with at least three SBP measurements during the study period were selected for the trajectory analyses. Data from eight

waves (i.e. 1989, 1991, 1997, 2000, 2004, 2006, 2009 and 2011) were included. The survey in 1993 was excluded from this study due to the lack of blood pressure measurements. The number of participants (%) were 3566 (100%) in 1989, 3206 (89.9%) in 1991, 2590 (72.6%) in 1997, 2834 (79.5%) in 2000, 2571 (72.1%) in 2004, 2564 (71.9%) in 2006, 2371 (66.5%) in 2009, and 2210 (62.0%) in 2011.

The study protocol of CHNS was approved by the institutional review board from the University of North Carolina at Chapel Hill and the National Institute for Nutrition and Food Safety, China Centre for Disease Control and Prevention. Written informed consent was collected from all participants.

## Data collection

During each wave, SBP was measured using a mercury sphygmomanometer on the right arm of each participant after a 10-minute seated rest [18]. SBP was measured three times at each visit, and the average of three measurements was used as the SBP level at each wave.

CVD status was determined based on self-reported information, which was available since the survey in 1997. CVDs were defined as having either myocardial infarction (MI) or stroke. The information of MI and stroke was retrieved based on the following questions: "Has the doctor ever given you the diagnosis of myocardial infarction?", and "Has the doctor ever given you the diagnosis of stroke?". Those giving a positive answer to either of the questions above were defined as having CVDs. Since the onset time of CVDs was not reported by the participants, we used the life-time cumulative risk of CVDs (i.e., ever report of having a diagnosis of MI or stroke during 1997–2011). Both recurrent and new cases of CVDs were included in the study.

The covariates included socio-demographic factors (e.g., age, sex, education, and living region), lifestyle behaviors (e.g., smoking, alcohol use, physical activity, and diet), use of anti-hypertensive medication, and body mass index (BMI). Data on socio-demographic factors, life-style behaviors, and use of anti-hypertensive medication were collected through self-reported questionnairres. Data on BMI was collected through physical examination. The socio-demographic factors, diet and BMI were collected at baseline (in 1989), however, data on smoking, alcohol intake and physical activity were used from the survey of 1997 because there was lack of these data at baseline.

Specifically, education was categorized according to the years of eduation: no formal school (i.e., <1 year), primary school (i.e., 1–6 years), middle school and above (i.e., >6 years). Living region was dichotomized into urban (i.e., living in a city) and rural (i.e., living in a village or town). Ever smoking was defined as a positive answer to the question "Have you ever smoked cigarettes or pipe?". High alcohol consumption was defined as drinking more than 3 times of alcoholic beverage per week regularly [19]. Physically inactive was defined as doing physical exercise less than 150-min of moderate physical activity or less than 75-min vigorous physical activity per week [20]. Data on dietary intake was collected from a questionnaire on 3-day records of household meals. Unfavorable diet was defined as having at least one of three macronutrients not meeting the dietary intake recommendation (i.e., 45–65% for carbohydrate, 20–35% for fat, and 10–35% for protein) [21]. Height and weight were measured, and BMI was calculated as measured weight (kg) divided by height (m) squared.

The use of anti-hypertensive medications was defined as the positive answer to the question "Are you currently taking anti-hypertensive drugs?". Regarding that the cumulative risk of CVDs during 1989–2011 was used as the outcome, the use of anti-hypertensive drugs was also identified as ever report of the positive answers during the study period.

## Statistical analyses

We performed latent class growth modeling (LCGM) to identify the SBP trajectories [22, 23]. Regarding that previous studies showed a nonlinear pattern of blood pressure trajectories [10, 24], so we considered two possible polynomial specifications (a linear and a quadratic) as a function of age to describe SBP trajectories. We used crude models to describe the SBP trajectories. For each of the polynomial models, one to seven class solution was described, starting with one-class linear model which assumed that all subjects follow the same trajectory over time (linear term); then, the number of latent classes and the polynomial term increases sequentially (S1 Table). The optimal number of classes was determined by the model with the lower Bayesian information criterion, high mean posterior class membership probabilities (>0.75), and significant level less than 0.05 of polynomial terms. We chose the final model of SBP trajectory also based on the assumption of SBP change over life-course (e.g., distinct change patterns including fluctuant variations).

The characteristics across trajectory groups were compared using ANOVA and chi-square tests for the continuous and categorical variables, respectively. Multiple imputation (5 times) was performed to impute the missing values of covariates (e.g., lifestyles and use of antihypertensive medications). Then, the analysis of association between SBP trajectory patterns and CVDs was performed through binary logistic regression model in the pooled dataset. Odds ratio (OR) and 95% confidence intervals (CI) were used to describe the associations from three models: model 1 was adjusted for age, sex, living region, education, and baseline SBP; model 2 was additionally adjusted for smoking, alcohol use, physical activities, diet, and BMI; and model 3 was further adjusted for use of anti-hypertensive drugs.

Two sensitivity analyses were performed. We examined the assocaition between SBP trajectory groups and CVDs after stratification by the use of anytihypertensive medication. In addition, to assess the effect of lost to follow-up, the association between SBP trajectory and CVDs was assessed among those who were followed from baseline until the last follow-up.

Stata 14.0 was used to perform the LCGM models and the associations, and R was used to plot the graph of life-course trajectories.

## Results

### The groups of SBP trajectory

We found five SBP trajectories according to different index of goodness of fit and discrimination (S1 Table). Fig 1 shows the five SBP trajectories over lifetime: Class 1 (red): rapid increase (n = 113, 3.2%); Class 2 (green): slight increase (n = 1958, 54.9%); Class 3 (blue): stable (n = 614, 17.2%); Class 4 (black): increase (n = 800, 22.4%); Class 5 (brown): fluctuant (n = 81, 2.3%). The "stable" group and "slight increase" group had similar patterns that SBP remained at a normal level (<140 mmHg) across adulthood, midlife and late life. The "increase" group was characterized by a normal SBP (<140 mmHg) from adulthood to midlife and a high SBP (≥140 mmHg) in late life. The group with "rapid increasing" had a more rapid increase in SBP with a normal SBP in adulthood and a high SBP during midlife and late life. The "fluctuant" group was characterized by a rapidly increasing SBP in adulthood and a decreasing SBP in midlife and late life. The SBP level in "fluctuant" group reached to the high level (≥140 mmHg) in adulthood and midlife and returned to normal level in late life.

### Characteristics across trajectory groups

In Table 1, across the five trajectory groups, the "stable" group were oldest, most likely to be women, and had highest proportion of people with no formal education (all $p<0.001$); the

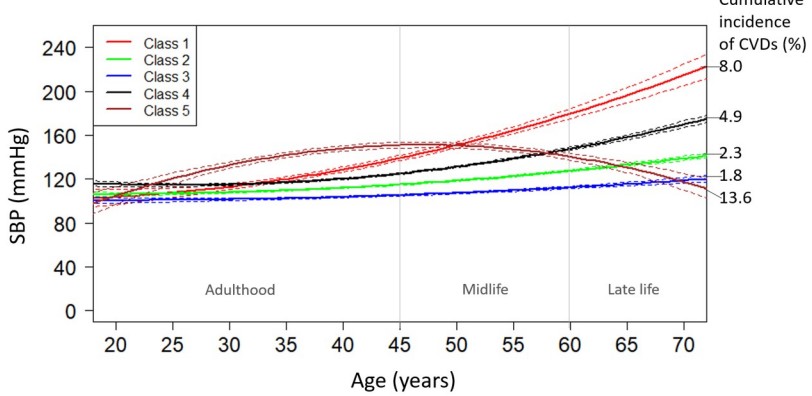

**Fig 1. Systolic blood pressure trajectories over lifetime.** Abbreviations: SBP = systolic blood pressure; CVDs = cardiovascular diseases. Class 1 (red): rapid increase (n = 113, 3.2%); Class 2 (green): slight increase (n = 1958, 54.9%); Class 3 (blue): stable (n = 614, 17.2%); Class 4 (black): increase (n = 800, 22.4%); Class 5 (brown): fluctuant (n = 81, 2.3%).

"fluctuant" group were most likely to live in urban area ($p = 0.009$), had highest prevalence of use of anti-hypertensive drugs, had highest level of BMI and SBP at baseline (all $p<0.001$); the "increase" group were most likely to smoke and drink alcohol (both $p<0.001$). There were no significant differences in the prevalence of physical inactivity ($p = 0.44$) and unfavorable diet ($p = 0.66$) across SBP trajectory groups.

## The association between SBP trajectories and CVDs

During the study period, 116 participants (3.2%) reported of having CVDs. After the adjustment of age, sex, living region, education, and baseline SBP, compared with "slight increase"

**Table 1. Characteristics of the study population across trajectory groups (n = 3566).**

| Characteristics* | Total | Class 1 (rapid increase) | Class 2 (slight increase) | Class 3 (stable) | Class 4 (increase) | Class 5 (fluctuant) | p |
|---|---|---|---|---|---|---|---|
| | (n = 3566) | (n = 113) | (n = 1958) | (n = 614) | (n = 800) | (n = 81) | |
| Age, years | 33.3 (6.8) | 32.4 (5.5) | 33.1 (6.8) | 35.5 (6.5) | 32.2 (6.6) | 33.9 (7.2) | <0.001 |
| Women | 1853 (52.0) | 63 (55.8) | 1019 (52.0) | 407 (66.3) | 320 (40.0) | 44 (54.3) | <0.001 |
| Urban | 975 (27.3) | 28 (24.8) | 542 (27.7) | 172 (28.0) | 198 (24.8) | 35 (43.2) | 0.009 |
| Education | | | | | | | |
| No formal school | 1014 (28.6) | 30 (26.5) | 541 (27.7) | 242 (39.7) | 174 (21.9) | 27 (33.3) | <0.001 |
| Primary school | 909 (25.6) | 30 (26.5) | 476 (24.4) | 156 (25.6) | 229 (28.8) | 18 (22.2) | |
| Middle school or above | 1627 (45.8) | 53 (46.9) | 934 (47.9) | 211 (34.6) | 393 (49.4) | 36 (44.4) | |
| Ever smoking | 1005 (34.5) | 25 (29.4) | 552 (34.5) | 133 (24.2) | 279 (45.1) | 16 (26.7) | <0.001 |
| High alcohol consumption | 507 (17.8) | 11 (12.9) | 281 (17.9) | 66 (12.2) | 142 (23.6) | 7 (11.9) | <0.001 |
| Physical inactivity | 2680 (91.4) | 77 (88.5) | 1484 (91.9) | 508 (92.0) | 556 (89.8) | 55 (91.7) | 0.44 |
| Unfavorable diet | 2666 (79.5) | 85 (81.7) | 1466 (79.1) | 474 (81.6) | 580 (78.7) | 61 (78.2) | 0.66 |
| Use of anti-hypertensive drugs | 498 (14.0) | 68 (60.2) | 130 (6.6) | 15 (2.4) | 229 (28.6) | 56 (69.1) | <0.001 |
| Body mass index, kg/ m$^2$ | 21.6 (2.4) | 22.5 (2.5) | 21.4 (2.3) | 20.9 (2.1) | 22.2 (2.5) | 23.6 (2.7) | <0.001 |
| Systolic blood pressure, mmHg | 111.1 (13.2) | 115.8 (11.6) | 109.6 (10.8) | 102.0 (10.9) | 118.3 (11.9) | 137.0 (21.1) | <0.001 |

Values are presented as mean (standard deviation) or n (%).

*The number of missing values was 16 for education, 652 for smoking, 710 for alcohol consumption, 634 for physical activity, 212 for diet, and 60 for body mass index. The missing value was imputed through multiple imputation for the subsequent analysis.

**Table 2. The association between systolic blood pressure trajectories and cardiovascular diseases (n = 3566).**

| Trajectory group | No. of subjects | No. of CVD cases | Odds ratio (95% confidence interval)* | | |
|---|---|---|---|---|---|
| | | | Model 1 | Model 2 | Model 3 |
| **Slight increase** | 1958 | 46 | Ref | Ref | Ref |
| **Stable** | 614 | 11 | 0.62 (0.31, 1.22) | 0.65 (0.32, 1.28) | 0.79 (0.40, 1.59) |
| **Increase** | 800 | 39 | 2.32 (1.46, 3.68) | 2.24 (1.40, 3.58) | 1.36 (0.82, 2.26) |
| **Rapid increase** | 113 | 9 | 4.40 (2.03, 9.51) | 3.95 (1.81, 8.62) | 1.68 (0.72, 3.89) |
| **Fluctuant** | 81 | 11 | 4.79 (1.95, 11.75) | 4.32 (1.76, 10.57) | 1.82 (0.71, 4.65) |

Abbreviations: CVD = cardiovascular disease.

*Model 1 was adjusted for socio-demographic factors (i.e., age, sex, living region and education) and baseline SBP, model 2 was further adjusted for smoking, high alcohol consumption, physical activity, unfavorable diet, and body mass index, and model 3 was additionally adjusted for antihypertensive drugs.

group, the OR (95% CI) of CVDs was 0.62 (0.31, 1.22) for "stable" group, 2.32 (1.46, 3.68) for "increase" group, 4.40 (2.03, 9.51) for "rapid increase" group, and 4.79 (1.95, 11.75) for "fluctuant" group (Table 2, model 1). The ORs slightly decreased after additional adjustment of lifestyles and BMI in model 2. After further adjustment of use of anti-hypertensive medication in model 3, the ORs were not significant in any of the trajectory groups.

The associations between SBP trajectory and CVDs stratified by the use of antihypertensive drugs were shown in S2 Table. Among those without using any antihypertensive drugs, there was no significant association between SBP trajectory groups and CVDs. Among those using antihypertensive drugs, compared with the "slight increase" group, the fully-adjusted OR was significant only for the "rapid increase" group, and the OR (95% CI) was 2.81 (1.01, 7.77).

## Discussion

### Main findings

Five distinct patterns of SBP trajectories from young adulthood to late life were identified: "stable", "slight increase", "increase", "rapid increase", and "fluctuant". Compared with the persons with "slight increase" SBP, those with "increase", "rapid increase", and "fluctuant" SBP over life-course had higher risk of CVDs. However, the antihypertensive drugs play a big role in the associations: only those antihypertensive drug users with "rapid increase" SBP over life-course had higher risk of CVDs.

### Comparison with previous studies on trajectory patterns

Since there are very few studies having assessed the life-course trajectory of SBP, it is difficult to directly compare our results to them. However, we can compare with the previous studies on blood pressure trajectory in specific life periods. Our study showed that the patterns of SBP trajectories differed in various age periods. For instance, there was an increasing trajectory of SBP (either gradually or rapidly) during adulthood before midlife, which was in line with the previous studies on trajectory in younger populations [11, 12]. Our findings also showed distinct SBP trajectories after midlife with an increasing, decreasing, or fluctuant pattern from midlife to old age, which is consistent with previous studies focusing on that part of life [13, 14]. The Rotterdam Study identified four SBP trajectories in an older population: gradual increase group, steeper increase group, decreasing group, and a group with modest variation [13], which was similar with our findings on SBP trajectory in late life. However, the findings in our study are notably a combination of the previous studies on blood pressure trajectory showing a bigger picture of SBP trajectory from a life-course perspective.

## Explanations of the associations between SBP trajectory and CVDs

The previous studies showed that the more rapid increase of blood pressure, the higher risk of CVDs in adulthood and midlife [10–12, 24, 25]. In line with them, we found a higher risk of CVDs in those with rapidly increasing SBP over lifetime. The possible explanation is that increase in blood pressure tends to cause rupture of an arteriole resulting in the high risk of CVDs (e.g., hemorrhagic stroke) [4]. In addition, the longer duration of hypertension could lead to stiffer vessels or arterial wall, which in turn leads to a resistance to anti-hypertensive medication and finally a higher risk of CVDs [26]. This can explain why the trajectory pattern with a rapid increase in SBP was associated with CVDs among those using antihypertensive medications. Our findings imply the importance of preventing rapid increase in SBP along with age. This is also pointed by a recent review on high blood pressure and CVDs, which showed that it is the time to pay greater attention to the prevention of the typical age-related increase in blood pressure in addition to the intensive treatment of established hypertension to eliminate the population burden of blood pressure related CVDs [27].

In addition, people with "fluctuant" SBP trajectory also had a significantly higher risk of CVDs, but the association disappeared after the adjustment of antihypertensive medications. This fluctuant group had the earliest onset of hypertension and the highest prevalence of using antihypertensive medications. The non-significant association with CVDs could be explained by the therapeutical effect of antihypertensive medications, which showed a decrease in SBP from high level to normal level during midlife and late life. This also emphasizes the need for effective treatment of hypertension and well control of blood pressure to reduce the risk of CVDs.

## Strengths and limitations

The major strengths of this study are the longitudinal cohort design, long follow-up period (>20 years), and repeated measurements of SBP which enable us to perform the trajectory analyses. Second, the large number of participants provided us enough statistical power to detect the distinct trajectory patterns. Third, the LCGM allows us to identify the heterogeneity of individual differences in SBP change, and it is more likely to find diverse trajectory which could be easily overlooked by other analysis.

However, the study has several limitations. First, the rate of lost to follow-up was quite high (from 10.1% in 1991 to 38% in 2011). However, the bias due to lost to follow-up might be minimized because the association between SBP trajectory and CVDs remained unchanged among those with the longest follow-up time (S3 Table). Second, statistical power might be reduced for the association between SBP trajectory and CVDs due to the small number of CVD cases especially in the stratified analyses. Third, the self-reported CVDs were used in the study, in which the results could be affected by the recall bias of these variables. Also, it is unable to differentiate the recurrent and new cases of CVDs. In addition, the results cannot rule out the cofounding effect from other factors (e.g., biomarkers) on CVDs due to lack of data.

## Conclusion

Having a rapidly increasing SBP over life-course is associated with a higher risk of CVDs. Our study implies the importance of lifetime monitoring the change of blood pressure for the prevention of CVDs.

## Supporting information

**S1 Table. The parameters of latent class growth models (n = 3566).** Abbreviations: BIC = Bayesian information criterion. *Model in bold font was chosen as the best model and

included as the main analysis in the study.
(DOCX)

**S2 Table. The association between systolic blood pressure trajectories and cardiovascular diseases by use of antihypertensive drugs (n = 3566).** Abbreviations: CVD = cardiovascular disease. *Model 1 was adjusted for socio-demographic factors (i.e., age, sex, living region and education) and baseline SBP, and model 2 was further adjusted for smoking, alcohol overconsumption, physical activity, unhealthy dietary and body mass index.
(DOCX)

**S3 Table. The association between systolic blood pressure trajectories and cardiovascular diseases among those followed from baseline to last visit (n = 2210).** Abbreviations: CVD = cardiovascular disease. *Model 1 was adjusted for socio-demographic factors (i.e., age, sex, living region and education) and baseline SBP, model 2 was further adjusted for smoking, alcohol overconsumption, physical activity, unhealthy dietary and body mass index, and model 3 was additionally adjusted for antihypertensive drugs.
(DOCX)

## Author Contributions

**Conceptualization:** Yongshi Xu, Jette Möller, Rui Wang, Yajun Liang.

**Formal analysis:** Yongshi Xu, Yajun Liang.

**Methodology:** Yongshi Xu, Yajun Liang.

**Software:** Yongshi Xu, Rui Wang.

**Supervision:** Jette Möller, Yajun Liang.

**Validation:** Rui Wang.

**Visualization:** Jette Möller, Rui Wang, Yajun Liang.

**Writing – original draft:** Yongshi Xu.

**Writing – review & editing:** Yongshi Xu, Jette Möller, Rui Wang, Yajun Liang.

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
