## [Decision Letter · Decision Letter 0]

18 Jun 2020

PONE-D-20-15275

Life-course blood pressure trajectories and cardiovascular diseases: a population-based cohort study in China

PLOS ONE

Dear Dr. Liang,

Thank you for submitting your manuscript to PLOS ONE. After careful consideration, we feel that it has merit but does not fully meet PLOS ONE’s publication criteria as it currently stands. Therefore, we invite you to submit a revised version of the manuscript that addresses the points raised during the review process.

Please revise your manuscript carefully according to the comments of the two Reviewers. Specifically, the classification of blood pressure trajectories might be revised, and be sure to make the data analyses valid.

We look forward to receiving your revised manuscript.

Kind regards,

Yan Li, MD, PhD

Academic Editor

PLOS ONE

Journal Requirements:

4. We note you have included a table to which you do not refer in the text of your manuscript. Please ensure that you refer to Table 1 in your text; if accepted, production will need this reference to link the reader to the Table.

Reviewers' comments:

Reviewer's Responses to Questions

**Comments to the Author**

1. Is the manuscript technically sound, and do the data support the conclusions?

Reviewer #1: Partly

Reviewer #2: Partly

2. Has the statistical analysis been performed appropriately and rigorously? 

Reviewer #1: No

Reviewer #2: Yes

3. Have the authors made all data underlying the findings in their manuscript fully available?

Reviewer #1: Yes

Reviewer #2: No

4. Is the manuscript presented in an intelligible fashion and written in standard English?

Reviewer #1: Yes

Reviewer #2: Yes

5. Review Comments to the Author

Reviewer #1: The authors identified five SBP blood pressure trajectories in a prospective cohort of 5144 Chinese people and found fluctuant SBP trajectory was associated with increased CV risk of non-lethal stroke and myocardial infarction in a subset of 3750 participants. There are some inertial problems that heavily weaken the credibility of the conclusion and should be properly addressed.

Major comments:

1. The SBP trajectories cannot be deemed as lifelong SBP trajectories. The follow-up duration is shorter compared to the whole lifespan and thus heterogenous participants, the youth and the elderly, are likely to be modelled into the same trajectory. Moreover 95% of total participants aged 17-47 years (Table 1). Modelling within this age range may be appropriate whereas modelling out of the range misleading. A clear trend toward persistent high blood pressure of the "fluctuant" group when compared to other groups is observed within this age range, which is consistent with our common sense that high blood pressure, but not "fluctuant" SBP, increases CV risk.

2. LCGM modelling strategy should be re-evaluated. It takes at least three repeated measures to conduct trajectory analysis, thus participants with 1-2 SBP readings should be excluded in the first place. Besides, each trajectory is expected to have at least 5% of the overall population. In the current analysis, "rapid increase" group and "fluctuant" group both comprise 2% of total participants, which make it hard to interpret at clinical perspective. Mean posterior probabilities are recommended at > 0.7-0.8, why did the authors choose 0.65 as the cutoff? In addition, the analysis of SBP trajectory and CV events is restricted to 3750 participants (73%), thus the BP trajectories should be re-modelled.

3. Since baseline SBP level is a strong predictor of prospective CV events and the authors had emphasized in the introduction that BP trajectory should be considered as an auxiliary factor to BP in CV risk evaluation, baseline SBP should be adjusted in the logistic model.

4. Given the proportion of loss to follow-up is high, and CV events within this study is self-reported, bias exists where a certain proportion of participants who had developed debilitating or lethal CVD, or died due to other cause, were neglected in the analyses due to the study design.

Minor comments:

1. Abstract: The OR and 95%CI of the "rapid increase" group is 2.10 (0.95, 4.64) which is non-significant. Thus it's not in support of the conclusion that "Rapid increase" patterns of SBP trajectory are associated with higher risk of CVDs.

2. Line 48-51: The authors expressed that, "BP change over time should be accounted in CVD risk estimation; monitoring BP with lifetime estimate is essential." The second sentence makes it hard to understand what the authors wanted to emphasize, the BP monitoring or the BP change over time, thus the sentences should be re-phrased.

Reviewer #2: Dr. Xu and colleagues investigated the association between life-course blood pressure trajectories and cardiovascular diseases in a population based cohort study in China. The major finding is that rapid increase and fluctuant patterns of SBP trajectory over life course are associated with higher risk of CVDs.

1. The study showed that the “fluctuant” group started with a low SBP (~70 mmHg) in childhood, which was not consistent with the common sense, please explain the reason.

2. Tables 2 and S2 showed different results between “Rapid increase” group and “Fluctuant” group, which needed to be further explained.

3. The table did not show the characteristics of the study population across trajectory groups (n=3750), which were analyzed in this study.

4. Did the events include both recurrent and new cases of CVD?

6. PLOS authors have the option to publish the peer review history of their article (what does this mean?). If published, this will include your full peer review and any attached files.

Reviewer #1: No

Reviewer #2: No

---

## [Author Response · Author response to Decision Letter 0]

8 Aug 2020

Journal Requirements:

Answer: We have revised the manuscript according to PLOS ONE style requirements.

Answer: The Data Availability statement has been changed and added in the cover letter. 

Answer: We have removed the phrase “data not shown”. All of the results are now shown in the revised manuscript. 

4. We note you have included a table to which you do not refer in the text of your manuscript. Please ensure that you refer to Table 1 in your text; if accepted, production will need this reference to link the reader to the Table.

Answer: We apologize for the carelessness. This has been revised and all figure and tables are now referred in the text. 

Answer: We have added the captions for the supporting materials after references, see pages 20-21, lines 400-403.

Reviewer #1: 

Major comments:

1. The SBP trajectories cannot be deemed as lifelong SBP trajectories. The follow-up duration is shorter compared to the whole lifespan and thus heterogenous participants, the youth and the elderly, are likely to be modelled into the same trajectory. Moreover 95% of total participants aged 17-47 years (Table 1). Modelling within this age range may be appropriate whereas modelling out of the range misleading. A clear trend toward persistent high blood pressure of the "fluctuant" group when compared to other groups is observed within this age range, which is consistent with our common sense that high blood pressure, but not "fluctuant" SBP, increases CV risk.

Answer: The age range of study participants was 9-70 years old at baseline. Thus, the 20-year’s SBP trajectory covered the age from 9 years old (youngest at baseline) to 90 years (oldest at the end of follow-up). The modelling of the trajectory was within this age range, which can be considered from a lifetime perspective. We agree that the trajectory in childhood and late life are less reliable due to the small number of participants. We have acknowledged this as a limitation in the revised version (see page 15, lines 274-275). 

2. LCGM modelling strategy should be re-evaluated. It takes at least three repeated measures to conduct trajectory analysis, thus participants with 1-2 SBP readings should be excluded in the first place. Besides, each trajectory is expected to have at least 5% of the overall population. In the current analysis, "rapid increase" group and "fluctuant" group both comprise 2% of total participants, which make it hard to interpret at clinical perspective. Mean posterior probabilities are recommended at > 0.7-0.8, why did the authors choose 0.65 as the cutoff? In addition, the analysis of SBP trajectory and CV events is restricted to 3750 participants (73%), thus the BP trajectories should be re-modelled.

Answer: We thank the reviewer for these suggestions. We have re-modelled the trajectory analysis among those with at least 3 SBP measurements (n=3600). Accordingly, the description and results in both abstract and main text have been revised (see pages 2,4,5).

In light of these analyses, we still choose the model with 5 trajectory classes as the final model. The mean posterior probability for each class in the updated model meets the recommended criteria (0.76-0.85). The percentage is still small for two groups: 2.3% for fluctuant group, and 3.1% for rapid increase group. We agree with the reviewer that each class is expected to have at least 5% of the participants, which is good to have enough power for the subsequent analysis. However, this is not a necessary requirement for a model decision as done in the previous studies (Buscot M-J, et al. Distinct child-to-adult body mass index trajectories are associated with different levels of adult cardiometabolic risk. Eur Heart J 2018; 39:2263-2270.). We acknowledged that with the small percent of some group (e.g., 3.1% for rapid increase), we have a reduced power for the subsequent associations between SBP trajectory and CVDs. We have acknowledged this as a limitation (see page 15, lines 275-278).

We know that statistical parameters (e.g., BIC, group membership, mean posterior probability) are very important for the model chosen, but this cannot be considered as the only criteria in the trajectory analysis. We have chosen the model also based on the assumption of SBP change over life course (e.g., distinct change patterns including fluctuant variations). If the model is chosen solely on the statistical parameters, we may not be able to identify those with fluctuant SBP levels, which do exist in reality and is informative from a clinical perspective. To make it more clear, we have modified the description in the statistical analysis (see page 7, lines 149-153). 

3. Since baseline SBP level is a strong predictor of prospective CV events and the authors had emphasized in the introduction that BP trajectory should be considered as an auxiliary factor to BP in CV risk evaluation, baseline SBP should be adjusted in the logistic model.

Answer: As the reviewer suggested, we have adjusted baseline SBP in the association between SBP trajectory and CVDs. The associations remain significant after additional adjustment of baseline SBP in model 1, and still remain significant for fluctuant and increase group in the fully-adjusted model. Accordingly, we revised the description in statistical analysis (see page 8, lines 160-161) and the notes under Table 2. 

4. Given the proportion of loss to follow-up is high, and CV events within this study is self-reported, bias exists where a certain proportion of participants who had developed debilitating or lethal CVD, or died due to other cause, were neglected in the analyses due to the study design.

Answer: In the study cohort, the rate of lost to follow-up ranged from 9.7% in 1991 to 36.4% in 2011. To minimize the bias due to lost to follow-up, we examined the cumulative risk of CVDs, which was defined as ever report of having CVDs during the study period. In the updated analysis, the information on CVDs were available from all of 3600 study participants in the updated analysis. Thus, the bias due to lost to follow-up for the outcome has been minimized in this study. 

Minor comments:

1. Abstract: The OR and 95%CI of the "rapid increase" group is 2.10 (0.95, 4.64) which is non-significant. Thus it's not in support of the conclusion that "Rapid increase" patterns of SBP trajectory are associated with higher risk of CVDs.

Answer: The abstract has been revised based on the new results, see pages 2-3. 

2. Line 48-51: The authors expressed that, "BP change over time should be accounted in CVD risk estimation; monitoring BP with lifetime estimate is essential." The second sentence makes it hard to understand what the authors wanted to emphasize, the BP monitoring or the BP change over time, thus the sentences should be re-phrased.

Answer: We have revised the sentence, see page 3, lines 48-50. 

Reviewer #2:

 1. The study showed that the “fluctuant” group started with a low SBP (~70 mmHg) in childhood, which was not consistent with the common sense, please explain the reason.

Answer: The SBP in childhood was estimated based on the model parameters. We agree that the low SBP of 70 mmHg is not common in reality. This could be due to the less reliable trajectory pattern in childhood because of the small number of participants in childhood. We have now acknowledged this as a limitation in the discussion (see page 15, lines 274-275). 

2. Tables 2 and S2 showed different results between “Rapid increase” group and “Fluctuant” group, which needed to be further explained.

Answer: We thank the reviewer for pointing this out. Now, the results have been updated based on the participants with 3 SBP measurements (n=3600) according to #1 Reviewer’s suggestion. The association between SBP trajectory and CVDs are shown in Table 2 and in Results (page 12, lines 199-207). 

3. The table did not show the characteristics of the study population across trajectory groups (n=3750), which were analyzed in this study.

Answer: We have updated the analysis in 3600 participants, which is the final sample with at least 3 SBP measurements and information on CVDs. The characteristics across trajectory groups are shown in Table 1 and Results (see page 9, lines 190-198).

4. Did the events include both recurrent and new cases of CVD?

Answer: We used self-reports of ever diagnosed CVDs as the outcome. The question is worded as follows: “Has a doctor ever given you the diagnosis of myocardial infarction (or stroke)?” Hence outcome included all events of CVDs. To make it clear, we have added the information in the methods section (see page 6, lines 108-109). We also revised the discussion to include the new and recurrent cases of CVDs (see page 15, lines 279-280).

---

## [Decision Letter · Decision Letter 1]

26 Aug 2020

PONE-D-20-15275R1

Life-course blood pressure trajectories and cardiovascular diseases: A population-based cohort study in China

PLOS ONE

Dear Dr. Liang,

Thank you for submitting your manuscript to PLOS ONE. After careful consideration, we feel that it has merit but does not fully meet PLOS ONE’s publication criteria as it currently stands. Therefore, we invite you to submit a revised version of the manuscript that addresses the points raised during the review process.

Please, consider the comments of the two Reviewers on the age range of the study population. Both Reviewers were concerned about the small number of subjects in the very young and older age groups.

We look forward to receiving your revised manuscript.

Kind regards,

Yan Li, MD, PhD

Academic Editor

PLOS ONE

Reviewers' comments:

Reviewer's Responses to Questions

**Comments to the Author**

1. If the authors have adequately addressed your comments raised in a previous round of review and you feel that this manuscript is now acceptable for publication, you may indicate that here to bypass the “Comments to the Author” section, enter your conflict of interest statement in the “Confidential to Editor” section, and submit your "Accept" recommendation.

Reviewer #1: (No Response)

Reviewer #2: (No Response)

2. Is the manuscript technically sound, and do the data support the conclusions?

Reviewer #1: Partly

Reviewer #2: Partly

3. Has the statistical analysis been performed appropriately and rigorously? 

Reviewer #1: Yes

Reviewer #2: Yes

4. Have the authors made all data underlying the findings in their manuscript fully available?

Reviewer #1: No

Reviewer #2: Yes

5. Is the manuscript presented in an intelligible fashion and written in standard English?

Reviewer #1: Yes

Reviewer #2: Yes

6. Review Comments to the Author

Reviewer #1: Three additional major comments should be addressed, on trajectory age range, use of antihypertensives and follow-up lost.

Re. authors' reply #1:

Baseline age for majority of participants falls in "adulthood" as illustrated in figure 1 and calculated from Table 1 (95% fall at 33.3±1.96*7.0 = 19.6-47.0 if normally distributed). After 20 years of follow-up, they turned 40-67 years ("midlife" and "late life"). Extreme young and old participants should be removed as they contributed to less than 5% of total population and trajectory at such age range was not reliable due to small sample size and lower-than-lifetime follow-up. Since removal of such participants would not change the result to a significant extent, the study will still offer information on blood pressure trajectory of participants from early-to-mid adulthood to mid-to-late adulthood, who are the major target for primary prevention of hypertension-related disorders.

The authors may partially rebut my comment by providing a table in the next revision of manuscript demonstrating that there is a consistent difference of CVD prevalence across blood pressure trajectories at different baseline age groups (e.g. grouped according to figure 1, baseline age: 9-18, 19-45, 46-60 and 61-70) , or, alternatively downplay the trajectory coverage, i.e. adulthood but not lifetime.

Interestingly, if the trajectories remain unchanged in the participants at adulthood at baseline after re-analysis, then adding of percentage of cardiovascular disease to the figure (see attachment) and limiting age range from 18-50 (black rectangle) will show us that participants at low level of blood pressure over adulthood had the lowest prevalence of cardiovascular disease and CVD prevalence increases with the increment of blood pressure, which is not surprising. Meanwhile, when limiting age range to 40-70 (red rectangle) then participants with highest CVD prevalence (class 4) had lower blood pressure. Since criteria of CVD were severe but non-lethal, then the result could be explained by the usage of antihypertensives for primary and secondary CVD prevention, that participants at class 2, 4 and 5 had significantly higher proportion of antihypertensive use (28%, 68% and 60%). That could also be confirmed from model 3 of table 2 that the significance nearly diminished after adjustment of antihypertensive treatment. A sensitivity analysis is needed to show if such association was still significant in antihypertensive drug-naïve subjects.

Re. authors' reply #2 & #3:

comments were properly addressed.

Re. authors' reply #4:

To minimize the impact of lost of follow-up due to various reasons, the authors should conduct another sensitivity analysis by re-evaluate their findings in those who participated the first and the last follow-up (i.e. 63.6% according to line 89).

Reviewer #2: Most comments were properly addressed. I have a minor suggestion regarding my previous comment 1:

Because of the small number of participants in childhood and the aged, the trajectory in childhood and later life shown in the figure might be less reliable. It is suggested that the age span in the analyses could be smaller, making the results more reliable.

7. PLOS authors have the option to publish the peer review history of their article (what does this mean?). If published, this will include your full peer review and any attached files.

Reviewer #1: No

Reviewer #2: No

---

## [Author Response · Author response to Decision Letter 1]

17 Sep 2020

Reviewer #1: 

Three additional major comments should be addressed, on trajectory age range, use of antihypertensives and follow-up lost.

Answer: We thank the reviewer very much for the further comments. We have addressed these three comments, please see the details of revisions as specified below. 

Re. authors' reply #1:

Baseline age for majority of participants falls in "adulthood" as illustrated in figure 1 and calculated from Table 1 (95% fall at 33.3±1.96*7.0 = 19.6-47.0 if normally distributed). After 20 years of follow-up, they turned 40-67 years ("midlife" and "late life"). Extreme young and old participants should be removed as they contributed to less than 5% of total population and trajectory at such age range was not reliable due to small sample size and lower-than-lifetime follow-up. Since removal of such participants would not change the result to a significant extent, the study will still offer information on blood pressure trajectory of participants from early-to-mid adulthood to mid-to-late adulthood, who are the major target for primary prevention of hypertension-related disorders.

The authors may partially rebut my comment by providing a table in the next revision of manuscript demonstrating that there is a consistent difference of CVD prevalence across blood pressure trajectories at different baseline age groups (e.g. grouped according to figure 1, baseline age: 9-18, 19-45, 46-60 and 61-70) , or, alternatively downplay the trajectory coverage, i.e. adulthood but not lifetime.

Answer: To address the reviewer’s comments, we have now removed the very young and very old participants by limiting the age range at baseline to 20-50 years (n=3566). We have revised the description of participants in the methods, see page 5. The trajectory pattern of SBP remain unchanged (see Fig 1 and S1 table). In addition, we updated the subsequent analysis on associations between SBP trajectories and cardiovascular diseases, see Table 1, S2-S3 tables and the text in the results (see pages 8-14). In addition, we have updated the abstract (see pages 2-3), main findings (see page 14) and discussion (see pages 15-17). 

Interestingly, if the trajectories remain unchanged in the participants at adulthood at baseline after re-analysis, then adding of percentage of cardiovascular disease to the figure (see attachment) and limiting age range from 18-50 (black rectangle) will show us that participants at low level of blood pressure over adulthood had the lowest prevalence of cardiovascular disease and CVD prevalence increases with the increment of blood pressure, which is not surprising. Meanwhile, when limiting age range to 40-70 (red rectangle) then participants with highest CVD prevalence (class 4) had lower blood pressure. Since criteria of CVD were severe but non-lethal, then the result could be explained by the usage of antihypertensives for primary and secondary CVD prevention, that participants at class 2, 4 and 5 had significantly higher proportion of antihypertensive use (28%, 68% and 60%). That could also be confirmed from model 3 of table 2 that the significance nearly diminished after adjustment of antihypertensive treatment. A sensitivity analysis is needed to show if such association was still significant in antihypertensive drug-naïve subjects.

Answer: As the reviewer suggested, we have added the cumulative incidence of CVDs to the figure (see Fig 1). We agree with reviewer that use of antihypertensive medication has an effect on the associations between SBP trajectory and CVDs. Now, we have added a supplementary table on the associations stratified by use of antihypertensive medications, see S2 table. Accordingly, we have added these results (see page 13) and a brief discussion (see page 16).

Re. authors' reply #4:

To minimize the impact of lost of follow-up due to various reasons, the authors should conduct another sensitivity analysis by re-evaluate their findings in those who participated the first and the last follow-up (i.e. 63.6% according to line 89).

Answer: As the reviewer suggested, we have performed the sensitivity analysis among those who were followed from baseline until the last follow-up (n=2210). The results were shown in S3 Table. The associations remained unchanged compared with the main results from total participants. Accordingly, we added a brief discussion on the lost of follow-up, see page 17. 

Reviewer #2:

Because of the small number of participants in childhood and the aged, the trajectory in childhood and later life shown in the figure might be less reliable. It is suggested that the age span in the analyses could be smaller, making the results more reliable.

Answer: We thank the reviewer for kindly providing further comments. As the reviewer suggested, we have removed the very young and very old participants (n=34) from baseline. The age range at baseline has been limited to 20-50 years (n=3566). We have described this in the methods, see page 5. Accordingly, we have updated the trajectory analysis (see Fig 1 and S1 Table) and subsequent analysis on associations between trajectory groups and cardiovascular diseases (see Table 1). Please also refer to our response to the first comments of Reviewer #1.

---

## [Decision Letter · Decision Letter 2]

5 Oct 2020

Life-course blood pressure trajectories and cardiovascular diseases: A population-based cohort study in China

PONE-D-20-15275R2

Dear Dr. Liang,

We’re pleased to inform you that your manuscript has been judged scientifically suitable for publication and will be formally accepted for publication once it meets all outstanding technical requirements.

Kind regards,

Yan Li, MD, PhD

Academic Editor

PLOS ONE

Additional Editor Comments (optional):

Reviewers' comments:

Reviewer's Responses to Questions

**Comments to the Author**

1. If the authors have adequately addressed your comments raised in a previous round of review and you feel that this manuscript is now acceptable for publication, you may indicate that here to bypass the “Comments to the Author” section, enter your conflict of interest statement in the “Confidential to Editor” section, and submit your "Accept" recommendation.

Reviewer #1: All comments have been addressed

Reviewer #2: All comments have been addressed

2. Is the manuscript technically sound, and do the data support the conclusions?

Reviewer #1: Yes

Reviewer #2: Yes

3. Has the statistical analysis been performed appropriately and rigorously? 

Reviewer #1: Yes

Reviewer #2: Yes

4. Have the authors made all data underlying the findings in their manuscript fully available?

Reviewer #1: Yes

Reviewer #2: Yes

5. Is the manuscript presented in an intelligible fashion and written in standard English?

Reviewer #1: Yes

Reviewer #2: Yes

6. Review Comments to the Author

Reviewer #1: (No Response)

Reviewer #2: (No Response)

7. PLOS authors have the option to publish the peer review history of their article (what does this mean?). If published, this will include your full peer review and any attached files.

Reviewer #1: No

Reviewer #2: No

---

## [Editor Report · Acceptance letter]

12 Oct 2020

PONE-D-20-15275R2 

Life-course blood pressure trajectories and cardiovascular diseases: A population-based cohort study in China 

Dear Dr. Liang:

I'm pleased to inform you that your manuscript has been deemed suitable for publication in PLOS ONE. Congratulations! Your manuscript is now with our production department. 

Kind regards, 

on behalf of

Professor Yan Li 

Academic Editor

PLOS ONE